# Sequential Causal Imitation Learning with Unobserved Confounders

**Daniel Kumor**
Purdue University
dkumor@purdue.edu

**Junzhe Zhang**
Columbia University
junzhez@cs.columbia.edu

**Elias Bareinboim**
Columbia University
eb@cs.columbia.edu

## Abstract

"Monkey see monkey do" is an age-old adage, referring to naïve imitation without a deep understanding of a system's underlying mechanics. Indeed, if a demonstrator has access to information unavailable to the imitator (monkey), such as a different set of sensors, then no matter how perfectly the imitator models its perceived environment (SEE), attempting to reproduce the demonstrator's behavior (DO) can lead to poor outcomes. Imitation learning in the presence of a mismatch between demonstrator and imitator has been studied in the literature under the rubric of causal imitation learning (Zhang et al., 2020), but existing solutions are limited to single-stage decision-making. This paper investigates the problem of causal imitation learning in sequential settings, where the imitator must make multiple decisions per episode. We develop a graphical criterion that is necessary and sufficient for determining the feasibility of causal imitation, providing conditions when an imitator can match a demonstrator's performance despite differing capabilities. Finally, we provide an efficient algorithm for determining imitability and corroborate our theory with simulations.

## 1 Introduction

Without access to observational data, an agent must learn how to operate at a suitable level of performance through trial and error (Sutton et al., 1998; Mnih et al., 2013). This from-scratch approach is often impractical in environments with the potential of extreme negative - and final - outcomes (driving off a cliff). While both Nature and machine learning researchers have approached the problem from a wide variety of perspectives, a particularly potent method which has been used with great success in many learning machines, including humans, is exploiting observations of other agents in the environment (Rizzolatti & Craighero, 2004; Hussein et al., 2017).

Learning to act by observing other agents offers a data multiplier, allowing agents to take into account others' experiences. Even when the precise loss function is unknown (what exactly goes into being a good driver?), the agent can attempt to learn from "experts", namely agents which are known to gain an acceptable reward at the target task. This approach has been studied under the umbrella of *imitation learning* (Argall et al., 2009; Billard et al., 2008; Hussein et al., 2017; Osa et al., 2018). Several methods have been proposed, including *inverse reinforcement learning* (Ng et al., 2000; Abbeel & Ng, 2004; Syed & Schapire, 2008; Ziebart et al., 2008) and *behavior cloning* (Widrow, 1964; Pomerleau, 1989; Muller et al., 2006; Mülling et al., 2013; Mahler & Goldberg, 2017). The former attempts to reconstruct the loss/reward function that the experts minimize and then use it for optimization; the latter directly copies the expert's actions (behavior cloning).

Despite the power entailed by this approach, it relies on a somewhat stringent condition: the expert and imitator's sensory capabilities need to be perfectly matched. As an example, self-driving cars rely solely on cameras or lidar, completely ignoring the auditory dimension - and yet most human demonstrators are able to exploit this data, especially in dangerous situations (car horns, screeching

35th Conference on Neural Information Processing Systems (NeurIPS 2021).

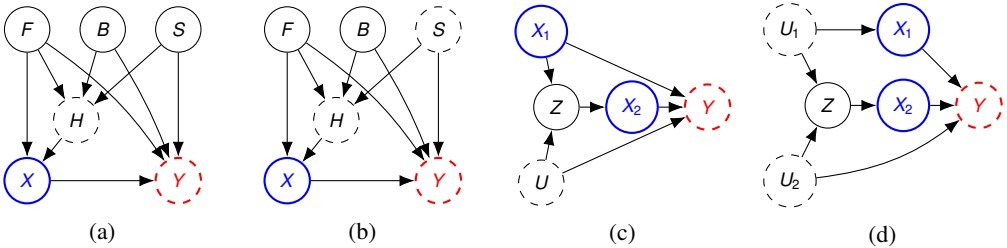

Figure 1: (a, b) represents a simplified view of a driver $X$ and surrounding cars $F, B, S$. (c) is imitable with policies $\pi_1(X_1) = P(X_1)$ and $\pi_2(X_2|Z) = P(X_2|Z)$, but in (d) $X_1, X_2$ is not imitable, despite there being a valid sequential backdoor.

tires). Perhaps without a microphone, the self-driving car would incorrectly attribute certain behaviors to visual stimuli, leading to a poor policy? For concreteness, consider the scenario shown in Fig. 1a, where the human driver ($X$, i.e., the demonstrator, in blue) is looking forward ($F$), and can hear car horns ($H$) from cars behind ($B$), and to the side ($S$). The driver's performance is represented by a variable $Y$ (red), which is unobserved (dashed node). Since our dataset only contains visual data, car horns $H$ remain unobserved to the learning agent (i.e., the imitator). Despite not being able to hear car horns, the learner from Fig. 1a had a full view of the car's surroundings, including cars behind and to the side, which turns out to be sufficient to perform imitation in this example. Consider an instance where $F, B, S$ are drawn uniformly over $\{0, 1\}$. The reward $Y$ is decided by $\neg X \oplus F \oplus B \oplus S$; $\oplus$ represents the *exclusive-or* operator. The human driver decides the action $X \leftarrow H$ where values of horn $H$ is given by $F \oplus B \oplus S$. Preliminary analysis reveals that the learner could perfectly mimic the demonstrator's decision-making process using an imitating policy $X \leftarrow F \oplus B \oplus S$. On the other hand, if the driving system does not have side cameras, the side view $S$ becomes latent; see Fig. 1b. The learner's reward $\mathbf{E}[Y|\mathrm{do}(\pi)]$ is equal to $0.5$ for any policy $\pi(x|f, b)$, which is far from the optimal demonstrator's performance, $\mathbf{E}[Y] = 1$.

Based on these examples, there arises the question of determining precise conditions under which an agent can account for the lack of knowledge or observations available to the expert, and how this knowledge should be combined to generate an optimal imitating policy, giving identical performance as the expert on measure $Y$. These questions have been recently investigated in the context of *causal imitation learning* (Zhang et al., 2020), where a complete graphical condition and algorithm were developed for determining imitability in the single-stage decision-making setting with partially observable models (i.e., in non-Markovian settings). Other structural assumptions, such as linearity (Etesami & Geiger, 2020), were also explored in the literature, but were still limited to a single action. Finally, de Haan et al. (2019) explore the case when expert and imitator can observe the same contexts, but the causal diagram is not available. Despite this progress, it is still unclear how to systematically imitate, or even whether imitation is possible when a learner must make several actions in sequence, where expert and imitator observe differing sets of variables (e.g., Figs. 1c and 1d).

The goal of this paper is to fill this gap in understanding. More specifically, our contributions are as follows. (1) We provide a graphical criterion for determining whether imitability is feasible in sequential settings based on a causal graph encoding the domain's causal structure. (2) We propose an efficient algorithm to determine imitability and to find the policy for each action that leads to proper imitation. (3) We prove that the proposed criterion is complete (i.e. both necessary and sufficient). Finally, we verify that our approach compares favorably with existing methods in contexts where a demonstrator has access to latent variables through simulations. Due to space constraints, proofs are provided in the complete technical report (Kumor et al., 2021).

## 1.1 Preliminaries

We start by introducing the notation and definitions used throughout the paper. In particular, we use capital letters for random variables ($Z$), and small letters for their values ($z$). Bolded letters represent sets of random variables and their samples ($\boldsymbol{Z} = \{Z_1, ..., Z_n\}$, $\boldsymbol{z} = \{z_1 \sim Z_1, ..., z_n \sim Z_n\}$). $|\boldsymbol{Z}|$ represents a set's cardinality. The joint distribution over variables $\boldsymbol{Z}$ is denoted by $P(\boldsymbol{Z})$. To simplify notation, we consistently use the shorthand $P(z_i)$ to represent probabilities $P(Z_i = z_i)$.

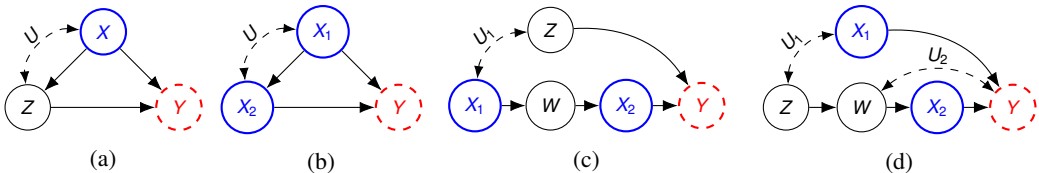

Figure 2: Despite there being no latent path between $Y$ and any $X$, the query in (a) is not imitable, but the query in (b) is imitable. While (c) is imitable if $Z$ comes before $X_2$ in temporal order, the query in (d) is imitable only if $Z$ comes before $X_1$.

The basic semantic framework of our analysis rests on *structural causal models* (SCMs) (Pearl, 2000, Ch. 7). An SCM $M$ is a tuple $\langle \boldsymbol{U}, \boldsymbol{V}, \boldsymbol{F}, P(\boldsymbol{u}) \rangle$ with $\boldsymbol{V}$ the set of endogenous, and $\boldsymbol{U}$ exogenous variables. $\boldsymbol{F}$ is a set of structural functions s.t. for $f_V \in \boldsymbol{F}$, $V \leftarrow f_V(pa_V, u_V)$, with $PA_V \subseteq \boldsymbol{V}, U_V \subseteq \boldsymbol{U}$. Values of $\boldsymbol{U}$ are drawn from an exogenous distribution $P(\boldsymbol{u})$, inducing distribution $P(\boldsymbol{V})$ over the endogenous $\boldsymbol{V}$. Since the learner can observe only a subset of endogenous variables, we split $\boldsymbol{V}$ into $\boldsymbol{O} \subseteq \boldsymbol{V}$ (observed) and $\boldsymbol{L} = \boldsymbol{V} \setminus \boldsymbol{O}$ (latent) sets of variables. The marginal $P(\boldsymbol{O})$ is thus referred to as the *observational distribution*.

Each SCM $M$ is associated with a causal diagram $\mathcal{G}$ where (e.g., see Fig. 2d) solid nodes represent observed variables $\boldsymbol{O}$, dashed nodes represent latent variables $\boldsymbol{L}$, and arrows represent the arguments $pa(V)$ of each functional relationship $f_V$. Exogenous variables $\boldsymbol{U}$ are not explicitly shown; a bi-directed arrow between nodes $V_i$ and $V_j$ indicates the presence of an unobserved confounder (UC) affecting both $V_i$ and $V_j$. We will use standard conventions to represent graphical relationships such as parents, children, descendants, and ancestors. For example, the set of parent nodes of $\boldsymbol{X}$ in $\mathcal{G}$ is denoted by $pa(\boldsymbol{X})_\mathcal{G} = \cup_{X \in \boldsymbol{X}} pa(X)_\mathcal{G}$. $ch$, $de$ and $an$ are similarly defined. Capitalized versions $Pa, Ch, De, An$ include the argument as well, e.g. $De(\boldsymbol{X})_\mathcal{G} = de(\boldsymbol{X})_\mathcal{G} \cup \boldsymbol{X}$. An observed variable $V_i \in \boldsymbol{O}$ is an *effective parent* of $V_j \in \boldsymbol{V}$ if there is a directed path from $V_i$ to $V_j$ in $\mathcal{G}$ such that every internal node on the path is in $\boldsymbol{L}$. We define $pa^+(\boldsymbol{S})$ as the set of effective parents of variables in $\boldsymbol{S}$, excluding $\boldsymbol{S}$ itself, and $Pa^+(\boldsymbol{S})$ as $\boldsymbol{S} \cup pa^+(\boldsymbol{S})$. Other relations, like $ch^+(\boldsymbol{S})$ are defined similarly.

A path from a node $X$ to a node $Y$ in $\mathcal{G}$ is said to be "active" conditioned on a (possibly empty) set $\boldsymbol{W}$ if there is a collider at $A$ along the path ($\rightarrow A \leftarrow$) only if $A \in An(\boldsymbol{W})$, and the path does not otherwise contain vertices from $\boldsymbol{W}$ (d-separation, Koller & Friedman (2009)). $\boldsymbol{X}$ and $\boldsymbol{Y}$ are independent conditioned on $\boldsymbol{W}$ $(\boldsymbol{X} \perp\!\!\!\perp \boldsymbol{Y}|\boldsymbol{W})_\mathcal{G}$ if there are no active paths between any $X \in \boldsymbol{X}$ and $Y \in \boldsymbol{Y}$. For a subset $\boldsymbol{X} \subseteq \boldsymbol{V}$, the subgraph obtained from $\mathcal{G}$ with edges outgoing from $\boldsymbol{X}$ / incoming into $\boldsymbol{X}$ removed is written $\mathcal{G}_{\underline{\boldsymbol{X}}}/\mathcal{G}_{\overline{\boldsymbol{X}}}$ respectively. Finally, we utilize a grouping of observed nodes, called *confounded components* (c-components, Tian & Pearl (2002); Tian (2002)).

**Definition 1.1.** *For a causal diagram $\mathcal{G}$, let $\boldsymbol{N}$ be a set of unobserved variables in $\boldsymbol{L} \cup \boldsymbol{U}$. A set $\boldsymbol{C} \subseteq Ch(\boldsymbol{N}) \cap \boldsymbol{O}$ is a **c-component** if for any pair $U_i, U_j \in \boldsymbol{N}$, there exists a path between $U_i$ and $U_j$ in $\mathcal{G}$ such that every observed node $V_k \in \boldsymbol{O}$ on the path is a collider (i.e., $\rightarrow V_k \leftarrow$).*

C-components correspond to observed variables whose values are affected by related sets of unobserved common causes, such that if $A, B \in \boldsymbol{C}$, $(A \not\perp\!\!\!\perp B|\boldsymbol{O} \setminus \{A, B\})$. In particular, we focus on *maximal* c-components $\boldsymbol{C}$, where there doesn't exist c-component $\boldsymbol{C}'$ s.t. $\boldsymbol{C} \subset \boldsymbol{C}'$. The collection of maximal c-components forms a partition $\boldsymbol{C}_1, \ldots, \boldsymbol{C}_m$ over observed variables $\boldsymbol{O}$. For any set $S \subseteq \boldsymbol{O}$, let $\boldsymbol{C}(S)$ be the union of c-components $\boldsymbol{C}_i$ that contain variables in $S$. For instance, for variable $Z$ in Fig. 1d, the c-component $\boldsymbol{C}(\{Z\}) = \{Z, X_1\}$.

## 2 Causal Sequential Imitation Learning

We are interested in learning a policy over a series of actions $\boldsymbol{X} \subseteq \boldsymbol{O}$ so that an imitator gets average reward $Y \in \boldsymbol{V}$ identical to that of an expert demonstrator. More specifically, let variables in $\boldsymbol{X}$ be ordered by $X_1, \ldots, X_n, n = |\boldsymbol{X}|$. Actions are taken sequentially by the imitator, where only information available at the time of the action can be used to inform a policy for $X_i \in \boldsymbol{X}$. To encode the ordering of observations and actions in time, we fix a topological ordering on the variables of $\mathcal{G}$, which we call the "temporal ordering". We define functions $\text{before}(X_i)$ and $\text{after}(X_i)$ to represent nodes that come before/after an action $X_i \in \boldsymbol{X}$ following the ordering, excluding $X_i$ itself. A policy $\pi$ on actions $\boldsymbol{X}$ is a sequence of decision rules $\{\pi_1, \ldots, \pi_n\}$ where each $\pi_i(X_i|\boldsymbol{Z}_i)$ is a function

mapping from domains of covariates $\boldsymbol{Z}_i \subseteq \text{before}(X_i)$ to the domain of action $X_i$. The imitator following a policy $\pi$ replacing the demonstrator in an environment is encoded by replacing the expert's original policy in the SCM $M$ with $\pi$, which gives the results of the imitator's actions as $P(\boldsymbol{V}|\text{do}(\pi))$. Our goal is to learn an imitating policy $\pi$ such that the induced distribution $P(Y|\text{do}(\pi))$ perfectly matches the original expert's performance $P(Y)$. Formally

**Definition 2.1.** *(Zhang et al., 2020) Given a causal diagram $\mathcal{G}$, $\boldsymbol{Y} \subseteq \boldsymbol{V}$ is said to be imitable with respect to actions $\boldsymbol{X} \subseteq \boldsymbol{O}$ in $\mathcal{G}$ if there exists $\pi \in \Pi$ uniquely discernible from the observational distribution $P(\boldsymbol{O})$ such that for all possible SCMs $M$ compatible with $\mathcal{G}$, $P(\boldsymbol{Y})_M = P(\boldsymbol{Y}|do(\pi))_M$.*

In other words, the expert's performance on reward $Y$ is imitable if any set of SCMs must share the same imitating policy $\pi \in \Pi$ whenever they generate the same causal diagram $\mathcal{G}$ and the observational distribution $P(\boldsymbol{O})$. Henceforth, we will consistently refer to Def. 2.1 as the *fundamental problem of causal imitation learning*. For single stage decision-making problems ($\boldsymbol{X} = \{X\}$), Zhang et al. (2020) demonstrated imitability for reward $Y$ if and only if there exists a set $\boldsymbol{Z} \subseteq \text{before}(X)$ such that $(Y \perp\!\!\!\perp X | \boldsymbol{Z})_{\mathcal{G}_{\underline{X}}}$, called the backdoor admissible set, (Pearl, 2000, Def. 3.3.1) ($\boldsymbol{Z} = \{F, B, S\}$ in Fig. 1a). It is verifiable that an imitating policy is given by $\pi(X|F, B, S) = P(X|F, B, S)$.

Since the backdoor criterion is complete for the single-stage problem, one may be tempted to surmise that a version of the criterion generalized to multiple interventions might likewise solve the imitability problem in the general case ($|\boldsymbol{X}| > 1$). Next we show that this is not the case. Let $\boldsymbol{X}_{1:i}$ stand for a sequence of variables $\{X_1, \ldots, X_i\}$; $\boldsymbol{X}_{1:i} = \emptyset$ if $i < 1$. Pearl & Robins (1995) generalized the backdoor criterion to the sequential decision-making setting as follows:

**Definition 2.2.** *(Pearl & Robins, 1995) Given a causal diagram $\mathcal{G}$, a set of action variables $\boldsymbol{X}$, and target node $Y$, sets $\boldsymbol{Z}_1 \subseteq \text{before}(X_1), \ldots, \boldsymbol{Z}_n \subseteq \text{before}(X_n)$ satisfy the sequential backdoor for $(\mathcal{G}, \boldsymbol{X}, Y)$ if for each $X_i \in \boldsymbol{X}$ such that $(Y \perp\!\!\!\perp X_i | \boldsymbol{X}_{1:i-1}, \boldsymbol{Z}_{1:i})_{\mathcal{G}_{\underline{X_i}\overline{\boldsymbol{x}}_{i+1:n}}}$.*

While the sequential backdoor is an extension of the backdoor to multi-stage decisions, its existence does not always guarantee the imitability of latent reward $Y$. As an example, consider the causal diagram $\mathcal{G}$ described in Fig. 1d. In this case, $\boldsymbol{Z}_1 = \{\}, \boldsymbol{Z}_2 = \{Z\}, \{(X_1, \boldsymbol{Z}_1), (X_2, \boldsymbol{Z}_2)\}$ is a sequential backdoor set for $(\mathcal{G}, \{X_1, X_2\}, Y)$, but there are distributions for which no agent can imitate the demonstrator's performance ($Y$) without knowledge of either the latent $U_1$ or $U_2$. To witness, suppose that the adversary sets up an SCM with binary variables as follows: $U_1, U_2 \sim Bern(0.5)$, with $X_1 := U_1$, $Z := U_1 \oplus U_2$, $X_2 := Z$ and $Y = \neg(X_1 \oplus X_2 \oplus U_2)$, with $\oplus$ as a binary XOR. The fact that $U \oplus U = 0$ is exploited to generate a chain where each latent variable appears exactly twice in $Y$, making $Y = \neg(U_1 \oplus (U_1 \oplus U_2) \oplus U_2) = 1$. On the other hand, when imitating, $X_1$ can no longer base its value on $U_1$, making the imitated $\hat{Y} = \neg(\hat{X}_1 \oplus \hat{X}_2 \oplus U_2)$. The imitator can do no better than $\mathbf{E}[\hat{Y}] = 0.5$! We refer readers to (Kumor et al., 2021, Proposition C.1) for a more detailed explanation.

## 2.1 Sequential Backdoor for Causal Imitation

We now introduce the main result of this paper: a generalized backdoor criterion that allows one to learn imitating policies in the sequential setting. For a sequence of covariate sets $\boldsymbol{Z}_1 \subseteq \text{before}(X_1), \ldots, \boldsymbol{Z}_n \subseteq \text{before}(X_n)$, let $\mathcal{G}'_i$, $i = 1, \ldots, n$, be the manipulated graph obtained from a causal diagram $\mathcal{G}$ by first (1) removing all arrows coming into nodes in $X_{i+1:n}$; and (2) adding arrows $\boldsymbol{Z}_{i+1} \to X_{i+1}, \ldots, \boldsymbol{Z}_n \to X_n$. We can then define a sequential backdoor criterion for causal imitation as follows:

**Definition 2.3.** *Given a causal diagram $\mathcal{G}$, a set of action variables $\boldsymbol{X}$, and target node $Y$, sets $\boldsymbol{Z}_1 \subseteq \text{before}(X_1), \ldots, \boldsymbol{Z}_n \subseteq \text{before}(X_n)$ satisfy the "**sequential $\pi$-backdoor**"[1] for $(\mathcal{G}, \boldsymbol{X}, Y)$ if at each $X_i \in \boldsymbol{X}$, either (1) $(X_i \perp\!\!\!\perp Y | \boldsymbol{Z}_i)$ in $(\mathcal{G}'_i)_{\underline{X_i}}$, or (2) $X_i \notin An(Y)$ in $\mathcal{G}'_i$.*

The first condition of Def. 2.3 is similar to the backdoor criterion where $\boldsymbol{Z}_i$ is a set of variables that effectively encodes all information relevant to imitating $X_i$ with respect to $Y$. In other words, if the joint $P(\boldsymbol{Z}_i \cup \{X_i\})$ matches when both expert and imitator are acting, then an adversarial $Y$ cannot distinguish between the two. The critical modification of the original $\pi$-backdoor for the sequential setting comes from the causal graph in which this check happens. $\mathcal{G}'_i$ can be seen as $\mathcal{G}$ with all future

---

[1]The $\pi$ in "$\pi$-backdoor" is part of the name, and does not refer to any specific policy.

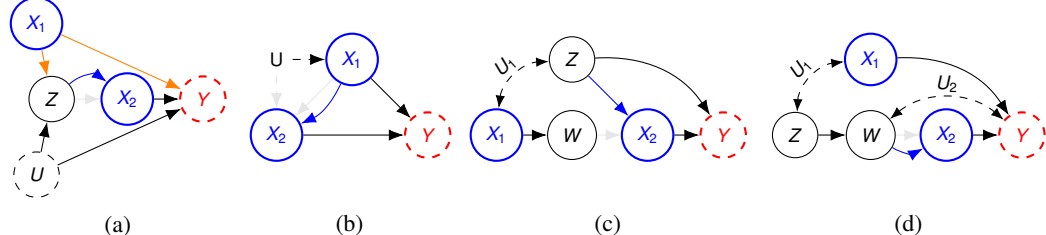

(a)            (b)            (c)            (d)

Figure 3: Examples of $\mathcal{G}'_1$. In Fig. 1c, we can have $\boldsymbol{Z}_1 = \emptyset, \boldsymbol{Z}_2 = \{Z\}$, so $X_2$ has its parents cut, and a new arrow added from $Z$ to $X_2$ (blue). The independence check $(X_1 \perp\!\!\!\perp Y | \emptyset)$ is done in graph (a) with edges outgoing from $X_1$ removed (orange). In Fig. 2b, using $\boldsymbol{Z}_1 = \emptyset, \boldsymbol{Z}_2 = \{X_1\}$, we first replace the parents of $X_2$ with just $X_1$ (b), and then remove both resulting outgoing edges from $X_1$ to check if $(X_1 \perp\!\!\!\perp Y)$. On the other hand, in Fig. 2c, if $\boldsymbol{Z}_2 = \{Z\}$, we get (c), which means $X_i \notin An(Y)$, passing condition 2 of Def. 2.3. Finally, in Fig. 2d, with $\boldsymbol{Z}_2 = \{W\}$, $X_1$ must condition on either $Z$ or $W$ to be independent of $Y$ in (d) once the edge $X_1 \to Y$ is removed.

actions of the imitator already encoded in the graph. That is, when performing a check for $X_i$, it is done with all actions after $i$ being performed by the imitator rather than expert, with the associated parents of each future $X_{j>i}$ replaced with their corresponding imitator's conditioning set. Several examples of $\mathcal{G}'_i$ are shown in Fig. 3.

The second condition allows for the case where an action at $X_i$ has no effect on the value of $Y$ once future actions are taken. Since $\mathcal{G}'_i$ has modified parents for future $\boldsymbol{X}_{j>i}$, the value of $X_i$ might no longer be relevant at all to $Y$, i.e. $Y$ would get the same input distribution no matter what policy is chosen for $X_i$. This allows $X_i$ to fail condition (1), meaning that it is not imitable by itself, but still be part of an imitable set $\boldsymbol{X}$, because future actions can shield $Y$ from errors made at $X_i$.

The distinction between condition 1 and condition 2 is shown in Fig. 3c: in the original graph $\mathcal{G}$ described in Fig. 2c, if $Z$ comes after $X_1$, then there is no valid adjustment set that can d-separate $X_1$ from $Y$. However, if the imitating policy for $X_2$ uses $Z$ instead of $W$ or $X_1$ (i.e. $\pi_{X_2} = P(X_2|Z)$), $X_1$ will no longer be an ancestor of $Y$ in $\mathcal{G}'_1$. In effect, the action made at $X_2$ ignores the inevitable mistakes made at $X_1$ due to not having access to confounder $U_1$ when taking the action.

Indeed, the sequential $\pi$-backdoor criterion can be seen as a recursively applying the single-action $\pi$-backdoor. Starting from the last action $X_k$ in temporal order, one can directly show that $Y$ is imitable using a backdoor admissible set $\boldsymbol{Z}_k$ (or $X_k$ doesn't affect $Y$ by any causal path). Replacing $X_k$ in the SCM with this new imitating policy, the resulting SCM with graph $G'_{k-1}$ has an identical distribution over $Y$ as $\mathcal{G}$. The procedure can then be repeated for $X_{k-1}$ using $G'_{k-1}$ as the starting graph, and continued recursively, showing imitability for the full set:

**Theorem 2.1.** *Given a causal diagram $\mathcal{G}$, a set of action variables $\boldsymbol{X}$, and target node $Y$, if there exist sets $\boldsymbol{Z}_1, \boldsymbol{Z}_2, ..., \boldsymbol{Z}_k$ that satisfy the sequential $\pi$-backdoor criterion with respect to $(\mathcal{G}, \boldsymbol{X}, Y)$, then $Y$ is imitable with respect to $\boldsymbol{X}$ in $\mathcal{G}$ with policy $\pi(X_i|\boldsymbol{Z_i}) = P(X_i|\boldsymbol{Z_i})$ for each $X_i \in \boldsymbol{X}$.*

Thm. 2.1 establishes the sufficiency of the sequential $\pi$-backdoor for imitation learning. Consider again the diagram in Fig. 2c. It is verifiable that covariate sets $\boldsymbol{Z}_1 = \{\}, \boldsymbol{Z}_2 = \{Z\}$ are sequential $\pi$-backdoor admissible. Thm. 2.1 implies that the imitating policy is given by $\pi_1(X_1) = P(X_1)$ and $\pi_2(X_2|Z) = P(X_2|Z)$. Once $\pi$-backdoor admissible sets are obtained, the imitating policy can be learned from the observational data through standard density estimation methods for stochastic policies, and supervised learning methods for deterministic policies. This means that the sequential $\pi$-backdoor is a method for choosing a set of covariates to use when performing imitation learning, which can be used instead of $Pa(X_i)$ for each $X_i \in \boldsymbol{X}$ in the case when the imitator does not observe certain elements of $Pa(\boldsymbol{X})$. With the covariates chosen using the sequential $\pi$-backdoor, one can use domain-specific algorithms for computing an imitating policy based on the observational data.

## 3   Finding Sequential $\pi$-Backdoor Admissible Sets

At each $X_i$, the sequential $\pi$-backdoor criterion requires that $\boldsymbol{Z}_i$ is a back-door adjustment set in the manipulated graph $\mathcal{G}'_i$. There already exist efficient methods for finding adjustment sets in the

literature (Tian & Paz, 1998; van der Zander & Liśkiewicz, 2020), so if the adjustment were with reference to $\mathcal{G}$, one could run these algorithms on each $X_i$ independently to find each backdoor admissible set $\boldsymbol{Z}_i$. However, each action $X_i$ has its $\boldsymbol{Z}_i$ in the manipulated graph $\mathcal{G}'_i$, which is *dependent on the adjustment used for future actions* $X_{i+1:n}$. This means that certain adjustment sets $\boldsymbol{Z}_j$ for $X_{j>i}$ will make there not exist any $\boldsymbol{Z}_i$ for $X_i$ in $\mathcal{G}'_i$ that satisfies the criterion! As an example, in Fig. 2c, $X_2$ can use any combination of $Z, X_1, W$ as a valid adjustment set $\boldsymbol{Z}_2$. However, if $Z$ comes after $X_1$ in temporal order, only $\boldsymbol{Z}_2 = \{Z\}$ leads to valid imitation over $\boldsymbol{X} = \{X_1, X_2\}$.

The direct approach towards solving this problem would involve enumerating all possible backdoor admissible sets $\boldsymbol{Z}_i$ for each $X_i$, but there are both exponentially many backdoor admissible sets $\boldsymbol{Z}_i$, and exponentially many combinations of sets over multiple $X_{i:n}$. Such a direct exponential enumeration is not feasible in practical settings. To address these issues, this section will see the development of Alg. 1, which finds a sequential $\pi$-backdoor admissible set $\boldsymbol{Z}_{1:n}$ with regard to actions $\boldsymbol{X}$ in a causal diagram $\mathcal{G}$ in polynomial time, if such a set exists.

Before delving into the details of Alg. 1, we describe a method intuitively motivated by the "Nearest Separator" from Van der Zander et al. (2015) that can generate a backdoor admissible set $\boldsymbol{Z}_i$ for a single independent action $X_i$ in the presence of unobserved variables. While it does not solve the problem of multiple actions due to the issues listed above, it is a building block for Alg. 1.

Consider the Markov Boundary (minimal Markov Blanket, Pearl (1988)) for a set of nodes $\boldsymbol{O}^X \subseteq \boldsymbol{O}$, which is defined as the minimal set $\boldsymbol{Z} \subset \boldsymbol{O} \setminus \boldsymbol{O}^X$ such that $(\boldsymbol{O}^X \perp\!\!\!\perp \boldsymbol{O} \setminus \boldsymbol{O}^X | \boldsymbol{Z})$. This definition can be applied to graphs with latent variables, where it can be constructed in terms of c-components:

**Lemma 3.1.** *Given $\boldsymbol{O}^X \subseteq \boldsymbol{O}$, the Markov Boundary of $\boldsymbol{O}^X$ in $\boldsymbol{G}$ is $Pa^+(\boldsymbol{C}(Ch^+(\boldsymbol{O}^X))) \setminus \boldsymbol{O}^X$*

If there is a set $\boldsymbol{Z} \subseteq \text{before}(X_i)$ that satisfies the backdoor criterion for $X_i$ with respect to $Y$, then taking $\mathcal{G}^Y$ as the ancestral graph of $Y$, the Markov Boundary $\boldsymbol{Z}'$ of $X_i$ in $\mathcal{G}^Y_{\underline{X}_i}$ has $\boldsymbol{Z}' \subseteq \text{before}(X_i)$, and also satisfies the backdoor criterion in $\mathcal{G}$ (Lem. C.1). The Markov Boundary can therefore be used to generate a backdoor adjustment set wherever one exists.

A naïve algorithm that uses the Markov Boundary of $X_i \in \boldsymbol{X}$ in $(\mathcal{G}'_i)^Y_{\underline{X}_i}$ as the corresponding $\boldsymbol{Z}_i$, and returns a failure whenever $\boldsymbol{Z}_i \notin \text{before}(X_i)$ for the sequential $\pi$-backdoor is equivalent to the existing literature on finding backdoor-admissible sets. It cannot create a valid sequential $\pi$-backdoor for Fig. 2c, since $X_2$ would have $\boldsymbol{Z}_2 = \{W\}$, but no adjustment set exists for $X_1$ that d-separates it from $Y$ in the resulting $\mathcal{G}'_1$. We must take into account interactions between actions encoded in $\mathcal{G}'_i$.

We notice that an $X_i$ does not require a valid adjustment set if it is not an ancestor of $Y$ in $\mathcal{G}'_i$ (i.e. $X_i$ does not need to satisfy (1) of Def. 2.3 if it can satisfy (2)). Furthermore, even if $X_i$ is an ancestor of $Y$ in $\mathcal{G}'_i$, and therefore must satisfy condition (1) of Def. 2.3, any elements of its c-component that are not ancestors of $Y$ in $\mathcal{G}'_i$ won't be part of $(\mathcal{G}'_i)^Y$, and therefore don't need to be conditioned.

It is therefore beneficial for an action $X_j$ to have a backdoor adjustment set that maximizes the number of nodes that are not ancestors of $Y$ in $\mathcal{G}'_{j-1}$, so that actions $X_{i<j}$ can satisfy (2) of Def. 2.3 if possible, and have the smallest possible c-components in $(\mathcal{G}'_i)^Y$ (increasing likelihood that backdoor set $\boldsymbol{Z}_i \subseteq \text{before}(X_i)$ exists if $X_i$ must satisfy condition (1)).

To demonstrate this intuition, we once again look at Fig. 2c, focusing only on action $X_2$. If we were to use $\{W\}$ as $\boldsymbol{Z}_2$, we still have the same set of ancestors of $Y$ in $\mathcal{G}'_1$. If we switch to $\{X_1\}$, then $W$ would no longer be an ancestor of $Y$ in $\mathcal{G}'_1$ - meaning that $X_1$ is *better* as a backdoor adjustment set for $X_2$ than $\{W\}$ if we only know that $X_2$ is an action (i.e. $W$ would directly satisfy (2) of Def. 2.3 if it were the other action). Finally, using $\{Z\}$ as $\boldsymbol{Z}_2$ makes both $X_1$ and $W$ no longer ancestors of $Y$ in $\mathcal{G}'_1$, meaning that it is the *best* option for the adjustment set $\boldsymbol{Z}_2$.

FINDOX in Alg. 1 employs the above ideas to iteratively grow a set $\boldsymbol{O}^X \subseteq \boldsymbol{O}$ of ancestors of $\boldsymbol{X}$ (and including $\boldsymbol{X}$) in $\mathcal{G}^Y$ whose elements (possibly excluding $\boldsymbol{X}$) will not be ancestors of $Y$ once the actions in their descendants are taken. That is, an element $O_i \in \boldsymbol{O}^X$ where $ch^+(O_i) \subset \boldsymbol{O}^X$ is not present in $(\mathcal{G}'_i)^Y$ for all actions $X_i$ that come before it in temporal order. Combined with the Markov Boundary, FINDOX can be used to generate sequential $\pi$-backdoors.

We exemplify the use of Alg. 1 through Fig. 2c. $\mathscr{O}^X$ represents a map of observed variables which are not ancestors of $Y$ in $\mathcal{G}'_{i<j}$ to the earliest action $X_j$ in their descendants. The keys of $\mathscr{O}^X$ will be the set $\boldsymbol{O}^X$. Considering the temporal order $\{X_1, Z, W, X_2, Y\}$, the algorithm starts from the last

---

**Algorithm 1** Find largest valid $O^X$ in ancestral graph of $Y$ given $\mathcal{G}$, $X$ and target $Y$

---

1: **function** HASVALIDADJUSTMENT($\mathcal{G}$,$O^X$,$O_i$,$X_i$)
2:     $C \leftarrow$ the c-component of $O_i$ in $\mathcal{G}^Y$
3:     $\mathcal{G}_C \leftarrow$ the subgraph of $\mathcal{G}^Y$ containing only $Pa^+(C)$ and intermediate latent variables
4:     $O^C \leftarrow C \setminus (O^X \cup \{O_i\})$ (elements of c-component that might be ancestors of $Y$ in $\mathcal{G}'_i$)
5:     **return** $\left(O_i \perp\!\!\!\perp O^C | (O^C \cap \text{before}(X_i))\right)$ in $\mathcal{G}_C$
6: **function** FINDOX($\mathcal{G}$,$X$,$Y$)
7:     $\mathscr{O}^X \leftarrow$ empty map from elements of $O$ to elements of $X$
8:     **do**
9:         **for** $O_i \in O$ of $\mathcal{G}^Y$ (ancestral graph of $Y$) in reverse temporal order **do**
10:             **if** $|ch^+(O_i)| > 0$ **and** $ch^+(O_i) \subseteq keys(\mathscr{O}^X)$ **then**
11:                $X_i \leftarrow$ earliest element of $\mathscr{O}^X[ch^+(O_i)]$ in temporal order
12:                **if** HASVALIDADJUSTMENT($\mathcal{G}$,$keys(\mathscr{O}^X)$,$O_i$,$X_i$) **then**
13:                    $\mathscr{O}^X[O_i] \leftarrow X_i$
14:             **else if** $O_i \in X$ and HASVALIDADJUSTMENT($\mathcal{G}$,$keys(\mathscr{O}^X)$,$O_i$,$O_i$) **then**
15:                $\mathscr{O}^X[O_i] \leftarrow O_i$
16:     **while** $|\mathscr{O}^X|$ changed in most recent pass
17:     **return** $keys(\mathscr{O}^X)$

---

node, $Y$, which has no children and is not an element of $X$, so is not added to $\mathscr{O}^X$. It then carries on to $X_2$, which is checked for the existence of a valid backdoor adjustment set. Here, the subgraph of the c-component of $X_2$ and its parents (HASVALIDADJUSTMENT) is simply $(W) \rightarrow (X_2)$, meaning that we can condition on $W$ to make $X_2$ independent of all other observed variables, including $Y$, in $\mathcal{G}_{\underline{X_2}}$ ($W$ is a Markov Boundary for $X_2$ in $\mathcal{G}_{\underline{X_2}}$). The algorithm therefore sets $\mathscr{O}^X = \{X_2 : X_2\}$, because $X_2$ is an action with a valid adjustment set. Notice that if the algorithm returned at this point with $O^X = \{X_2\}$, the Markov Boundary of $O^X$ in $\mathcal{G}_{\underline{X_2}}$ is $W$, and corresponds to a sequential $\pi$-backdoor for the single action $X_2$ (ignoring $X_1$), with policy $\pi(X_2|W) = P(X_2|W)$.

Next, $W$ has $X_2$ as its only child, which itself maps to $X_2$ in $\mathscr{O}^X$. The subgraph of $W$'s c-component and its parents is $(X_1) \rightarrow (W)$, giving $(W \perp\!\!\!\perp O|X_1)_{\mathcal{G}_W}$, and $\{X_1\} \subseteq \text{before}(X_2)$, allowing us to conclude that there is a backdoor admissible set for $X_2$ where $W$ is no longer an ancestor of $Y$. We set $\mathscr{O}^X = \{X_2 : X_2, W : X_2\}$, and indeed with $O^X = \{X_2, W\}$, the Markov Boundary of $O^X$ in $\mathcal{G}_{\underline{X_2}}$ is $X_1$, and is once again a valid sequential $\pi$-backdoor for the single action $X_2$ (ignoring $X_1$), with policy $\pi(X_2|X_1) = P(X_2|X_1)$. The $W$ in $O^X$ was correctly labeled as not being an ancestor of $Y$ after action $X_2$ is taken.

Since $Z$ doesn't have its children in the keys of $\mathscr{O}^X$, and is not an element of $X$, it is skipped, leaving only $X_1$. $X_1$'s children ($W$) are in $\mathscr{O}^X$, we check conditioning using $X_2$ instead of $X_1$ (i.e. we check if $X_1$ can satisfy (2) of Def. 2.3, and not be an ancestor of $Y$ in $\mathcal{G}'_1$). This time, we have $(X_1) \leftrightarrow (Z)$ as the c-component subgraph, and $Z$ comes before $X_2$, satisfying the check $(X_1 \perp\!\!\!\perp Z|Z)$ in HASVALIDADJUSTMENT, resulting in $\mathscr{O}^X = \{X_2 : X_2, W : X_2, X_1 : X_2\}$, and $O^X = \{X_2, W, X_1\}$. Indeed, the Markov Boundary of $O^X$ in $\mathcal{G}_{\underline{X_2}}$ is $\{Z\}$, and we can construct a valid sequential $\pi$-backdoor by using $Z_1 = \{\}$ and $Z_2 = \{Z\}$, where $X_1$ is no longer an ancestor of $Y$ in $\mathcal{G}'_1$! In this case, we call $X_2$ a "boundary action", because it is an ancestor of $Y$ in $\mathcal{G}'_2$. On the other hand, $X_1$ is not a boundary action, because it is not an ancestor of $Y$ in $\mathcal{G}'_1$.

**Definition 3.1.** *The set $X^B \subseteq X$ called the "boundary actions" for $O^X := \text{FINDOX}(\mathcal{G}, X, Y)$ are all elements $X_i \in X \cap O^X$ where $ch^+(X_i) \not\subseteq O^X$.*

Alg. 1 is general: the set $O^X$ returned by FINDOX can always be used in conjunction with its Markov Boundary to construct a sequential $\pi$-backdoor if one exists:

**Lemma 3.2.** *Let $O^X := \text{FINDOX}(\mathcal{G}, X, Y)$, and $X' := O^X \cap X$. Taking $Z$ as the Markov Boundary of $O^X$ in $\mathcal{G}^Y_{\underline{X'}}$, and $X^B$ as the boundary actions of $O^X$, the sets $Z_i = (Z \cup X^B) \cap \text{before}(X'_i)$ for each $X'_i \in X'$ are a valid sequential $\pi$-backdoor for $(\mathcal{G}, X', Y)$.*

**Lemma 3.3.** *Let $O^X := \text{FINDOX}(\mathcal{G}, X, Y)$. Suppose that there exists a sequential $\pi$-backdoor for $X" \subseteq X$. Then $X" \subseteq O^X$.*

Combined together, Lems. 3.2 and 3.3 show that FINDOX finds the *maximal* subset of $X$ where a sequential $\pi$-backdoor exists, and the adjustment sets $Z_{1:n}$ can be constructed using the subset of a Markov Boundary over $O^X$ that comes before each corresponding action $X_i$ (Lem. 3.2). FINDOX is therefore both necessary and sufficient for generating a sequential $\pi$-backdoor:

**Theorem 3.1.** *Let $O^X$ be the output of $\text{FINDOX}(\mathcal{G}, X, Y)$. A sequential $\pi$-backdoor exists for $(\mathcal{G}, X, Y)$ if and only if $X \subseteq O^X$.*

## 4 Necessity of Sequential $\pi$-Backdoor for Imitation

In this section, we show that the sequential $\pi$-backdoor is *necessary* for imitability, meaning that the sequential $\pi$-backdoor is complete.

A given imitation problem can have multiple possible conditioning sets satisfying the sequential $\pi$-backdoor, and a violation of the criterion for one set does not preclude the existence of another that satisfies the criterion. To avoid this issue, we will use the output of Algorithm FINDOX, which returns a unique set $O^X$ for each problem:

**Lemma 4.1.** *Let $O^X := \text{FINDOX}(\mathcal{G}, X, Y)$. Suppose $\exists X_i \in X$ s.t. $X_i \in X \setminus O^X$. Then $X$ is not imitable with respect to $Y$ in $\mathcal{G}$.*

Our next proposition establishes the necessity of the sequential $\pi$-backdoor criterion for the imitability of the expert's performance (Def. 2.1), which follows immediately from Lem. 4.1 and Thm. 3.1.

**Theorem 4.1.** *If there do not exist adjustment sets satisfying the sequential $\pi$-backdoor criterion for $(\mathcal{G}, X, Y)$, then $X$ is not imitable with respect to $Y$ in $\mathcal{G}$.*

The proof of Lem. 4.1 relies on the construction of an adversarial SCM for which $Y$ can detect the imitator's lack of access to the latent variables. For example, in Fig. 2a, $Z$ can carry information about the latent variable $U$ to $Y$, and is only determined after the decision for the value of $X$ is made. Setting $U \sim Bern(0.5), X := U, Z := U, Y := X \oplus Z$ leaves the imitator with a performance of $\mathbf{E}[\hat{Y}] = 0.5$, while the expert can get perfect performance ($\mathbf{E}[Y] = 1$).

Another example with similar mechanics can be seen in Fig. 2c. If the variables are determined in the order $(X_1, W, X_2, Z, Y)$, then the sequence of actions is not imitable, since $Z$ can transfer information about the latent variable $U$ to $Y$, while $X_2$ has no way of gaining information about $U$, because the action at $X$ needed to be taken without context.

Finally, observe Fig. 2d. If $Z$ is determined *after* $X_1$, the imitator must guess a value for $X_1$ without this side information, which is then combined with $U_2$ at $W$. An adversary can exploit this to construct a distribution where guessing wrong can be detected at $Y$ as follows: $U_1 \sim Bern(0.5)$, $Z, X := U_1, U_2 \sim (Bern(0.5), Bern(0.5))$ (that is, $U_2$ is a tuple of two binary variables, or a single variable with a uniform domain of $0, 1, 2, 3$). Then setting $W = U_2[Z]$ ([] represents array access, meaning first element of tuple if $Z = 0$ and second if $Z = 1$), and $X_2 := W, Y := (U_2[X_1] == X_2)$ gives $\mathbf{E}[Y] = 1$ only if $\pi_1$ guesses the value of $U_1$, meaning that the imitator can never achieve the expert's performance. This construction also demonstrates non-imitability when $X_1$ and $Z$ are switched (i.e., Fig. 2c with $W \leftrightarrow Y$ added, and $X_1$ coming before $Z$ in temporal order).

Due to these results, after running Alg. 1 on the domain's causal structure, the imitator gets two pieces of information:

1. Is the problem imitable? In other words, is it possible to use only observable context variables, and still get provably optimal imitation, despite the expert and imitator having different information?

2. If so, what context should be included in each action? Including/removing certain observed covariates in an estimation procedure can lead to different conclusions/actions, only one of which is correct (known as "Simpson's Paradox" in the statistics literature (Pearl, 2000)). Furthermore, as demonstrated in Fig. 2c, when performing actions sequentially, some actions might not be imitable themselves ($X_1$ if $Z$ after $X_1$), which leads to bias in observed

| # | Structure | Order | Seq. $\pi$-Backdoor | $\pi$-Backdoor | Observed Parents | All Observed |
|---|---|---|---|---|---|---|
| 1 | | $Z, X_1,$ $X_2, Y$ | $0.04 \pm 0.04\%$ | $0.04 \pm 0.03\%$ | $0.05 \pm 0.04\%$ | **$0.13 \pm 0.18\%$** |
| 2 | | $Z, X_1,$ $X_2, Y$ | $0.05 \pm 0.03\%$ | $0.05 \pm 0.03\%$ | **$0.20 \pm 0.25\%$** | $0.05 \pm 0.03\%$ |
| 3 | | $X_1, Z,$ $X_2, Y$ | $0.04 \pm 0.03\%$ | **Not Imitable** | **$0.27 \pm 0.40\%$** | **$0.26 \pm 0.39\%$** |
| 4 | | $X_1, Z,$ $X_2, Y$ | Not Imitable | Not Imitable | **$0.19 \pm 0.29\%$** | **$0.19 \pm 0.29\%$** |

Table 1: Values of $|\mathbf{E}[Y] - \mathbf{E}[\hat{Y}]|$ from behavioral cloning using different contexts in randomly sampled models consistent with each causal graph. Cases with incorrect imitation are shown in red.

descendants ($W$) - the correct context takes this into account, using only covariates known not to be affected by incorrectly guessed actions.

Finally, the obtained context $\boldsymbol{Z}_i$ for every action $X_i$ could be be used as input to existing algorithms for behavioral cloning, giving an imitating policy with an unbiased result.

# 5 Simulations

We performed 2 experiments (for full details, refer to Kumor et al. (2021, Appendix B)), comparing the performance of 4 separate approaches to determining which variables to include in an imitating policy:

1. **All Observed (AO)** - Take into account all variables available to the imitator at the time of each action. This is the approach most commonly used in the literature.

2. **Observed Parents (OP)** - The expert used a set of variables to take an action - use the subset of these that are available to the imitator.

3. **$\pi$-Backdoor** - In certain cases, each individual action can be imitated independently, so the individual single-action covariate sets are used.

4. **Sequential $\pi$-Backdoor (ours)** - The method developed in this paper, which takes into account multiple actions in sequence.

The first simulation consists of running behavioral cloning on randomly sampled distributions consistent with a series of causal graphs designed to showcase aspects of our method. For each causal graph, 10,000 random discrete causal models were sampled, representing the environment as well as expert performance, and then the expert's policy $\boldsymbol{X}$ was replaced with imitating policies approximating $\pi(X_i) = P(X_i|ctx(X_i))$, with context $ctx$ determined by each of the 4 tested methods in turn. Our results are shown in Table 1, with causal graphs shown in the first column, temporal ordering of variables in the second column, and absolute distance between expert and imitator for the 4 methods in the remaining columns.

In the first row, including $Z$ when developing a policy for $\boldsymbol{X}$ leads to a biased answer, which makes the average error of using all observed covariates (red) larger than just the sampling fluctuations present in the other columns. Similarly, $Z$ needs to be taken into account in row 2, but

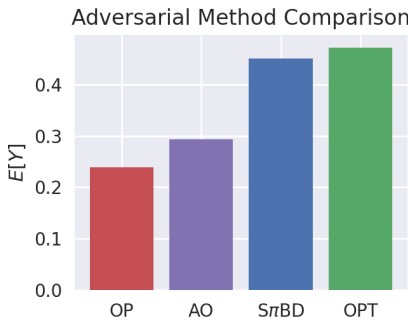

Figure 4: Results of applying supervised learning techniques to continuous data with different sets of variables as input at each action. OPT is the ground truth expert's performance, S$\pi$BD represents our method, AO is all observed, and OP represents observed parents.

it is not explicitly used by $X$, so a method relying only on observed parents leads to bias here. In the next row, $Z$ is not observed at the time of action $X_1$, making the $\pi$-backdoor incorrectly claim non-imitability. Our method recognizes that $X_2$'s policy can fix the error made at $X_1$, and is the only method that leads to an unbiased result. Finally, in the 4th row, the non-causal approaches have no way to determine non-imitability, and return biased results in all such cases.

The second simulation used a synthetic, adversarial causal model, enriched with continuous data from the HighD dataset (Krajewski et al., 2018) altered to conform to the causal model, to demonstrate that different covariate sets can lead to significantly different imitation performance. A neural network was trained for each action-policy pair using standard supervised learning approaches, leading to the results shown in Fig. 4. The causal structure was not imitable from the single-action setting, so the remaining 3 methods were compared to the optimal reward, showing that our method approaches the performance of the expert, whereas non-causal methods lead to biased results. Full details of model construction, including the full causal graph are given in (Kumor et al., 2021, Appendix B)

## 6 Limitations & Societal Impact

There are two main limitations to our approach: (1) Our method focuses on the causal diagram, requiring the imitator to provide the causal structure of its environment. This is a fundamental requirement: any agent wishing to operate in environments with latent variables must somehow encode the additional knowledge required to make such inferences from observations. (2) Our criterion only takes into consideration the causal structure, and not the associated data $P(o)$. Data-dependent methods can be computationally intensive, often requiring density estimation. If our approach returns "imitable", then the resulting policies are guaranteed to give perfect imitation, without needing to process large datasets to determine imitability.

Finally, advances in technology towards improving imitation can easily be transferred to methods used for impersonation - our method provides conditions under which an imposter (imitator) can fool a target ($Y$) into believing they are interacting with a known party (expert). Our method shows when it is provably impossible to detect an impersonation attack. On the other hand, our results can be used to ensure that the causal structure of a domain cannot be imitated, helping mitigate such issues.

## 7 Conclusion

Great care needs to be taken in choosing which covariates to include when determining a policy for imitating an expert demonstrator when expert and imitator have different views of the world. The wrong set of variables can lead to biased, or even outright incorrect predictions. Our work provides general and complete results for the graphical conditions under which behavioral cloning is possible, and provides an agent with the tools needed to determine the variables relevant to its policy.

### Funding Transparency

The authors were partially supported by grants from NSF IIS-1704352 and IIS-1750807 (CAREER). Daniel Kumor also acknowledges additional revenue from an internship at Amazon.

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
