# OpenReview forum: "Sequential Causal Imitation Learning with Unobserved Confounders"
_NeurIPS.cc/2021/Conference — NeurIPS 2021 Oral_

### Official Review · Reviewer_1SfX · 2021-07-10

**Rating:** 7
**Confidence:** 3

**Summary:**

This paper studies the problem of imitation learning when the expert and learner differ in their available information and input. The authors extend the work of Zhang e al., 2020 from single stage to the sequential setting. A necessary and sufficient graphical criterion is proposed to determine if optimal imitation (P(y | do(pi)) = P(y)) is possible. An algorithm to determine the set of covariates enabling such imitation is also proposed. Experiments showcasing the value of the new approach are conducted on synthetic and semi-synthetic data.

**Limitations And Societal Impact:**

The authors have addressed them well.

**Main Review:**

The strengths of this paper are the nice theoretical extensions to  Zhang e al., 2020. Naively applying  Zhang et al., 2020 to the sequential case by considering each sequential action independently may incorrectly prescribe “not-imitable” to some graphs, The new graphical criterion in this work considers all sequential actions in conjunction (using similar flavoured techniques as sequential backdoor) to give the correct result. The algorithm to determine which covariates to use is also useful and surely will lend itself to new applications.

The first weakness of this paper is strong assumptions needed, thus limiting its practicality. The main concern of the paper is the very strong imitability criterion P(y | do(pi)) = P(y). In practice we may be more concerned with simply maximizing E[y] even if a graph is not truly imitable. However because this work is data independent, this is something that cannot be handled.

Another strong assumption is that the full causal graph has to be known and specified a priori. While this assumption is probably very difficult to remove completely, it could be taken into consideration by providing error bounds for mismatched causal graphs, etc.

I also wish that the causal graphs studied were more realistic. For example, the HighD experiment’s example of air conditioning being the collider for mental state and road conditions feels a bit artificial. I understand that the experiment is largely meant to be illustrative, but it is nonetheless a weakness. Given that the method requires the full causal graph, it is more likely that practitioners will come up with more simple and structured graphs rather than intricate and detailed ones. For example, it may be more practical if the authors could have studied graphs of particular structure (e.g. block MDPs), which seems to be more likely to be used in the real world. Likewise, the current analysis requires different covariates for each step, with no clear solutions for infinite (or long horizon) settings with some repeated structure, which seem to be the predominant setting in the real world.

Despite deferring the bulk of the proofs to the appendix, the paper is still quite dense due to its heavy use of inline notation. But given the nature of the work I’m not sure how avoidable it is. Perhaps using more of the high level examples (sound of horn, etc.) in place of the later examples of SCMs with xor functions, etc. could make the paper more approachable.

Overall I feel the paper makes some solid contributions, and will update my score if some of the concerns can be addressed.

---
**EDIT**: I have updated my score, for more details see the reply to the author's response.


**Time Spent Reviewing:**

2

---

> ### Author Response · Authors · 2021-08-10
> **Response to Reviewer 1SfX**
>
> We are glad that our algorithm was found useful, and that our paper was considered a solid contribution. We hope to assuage the mentioned concerns in the sequel and ask you to reevaluate the contributions of the paper.  Still, we would be happy to provide further elaboration  if you find some issues were not answered satisfactorily.
>
> ---
> > _“The main concern of the paper is the very strong imitability criterion P(y | do(pi)) = P(y). In practice we may be more concerned with simply maximizing E[y] even if a graph is not truly imitable.”_
>
> We respectfully disagree with the statement that the imitability criterion $P(y|do(\pi)) = P(y)$ is overly strong and “In practice we may be more concerned with simply maximizing $E[y]$”.
>
> First, we note that oftentimes the reward signal $Y$ is unobserved and the reward function is not specified (Ng & Russell, 2000). In this case, Corollary 1 in (Zhang et al., 2020) showed that the expected reward $E[Y|do(\pi)]$ of any candidate policy is not identifiable. That is, it is generally infeasible to estimate the expected reward $E[Y|do(\pi)]$ from the observational data, let alone finding an optimal policy $\pi^*$ that maximizes $E[Y|do(\pi)]$.
>
> A common approach to circumvent this issue found throughout the literature is to assume that the expert achieves near optimal performance $E[Y]$, which leads to the imitation learning paradigm (Ng & Russell, 2000). Our goal of in this paper is to find an imitating policy $\pi$ that satisfies the equality condition $P(y|do(\pi)) = P(y)$. If such condition holds, the performance of the imitator $E[Y|do(\pi)]$ must match that of the expert $E[Y]$ and, therefore, is also satisfactory. In other words, we are able to obtain an effective policy by training an imitator that perfectly mimics the expert’s performance, despite of being unable to directly maximizing the expected reward $E[Y|do(\pi)]$.
>
> ---
> > _“Another strong assumption is that the full causal graph has to be known and specified a priori. While this assumption is probably very difficult to remove completely, it could be taken into consideration by providing error bounds for mismatched causal graphs, etc.”_
>
> Unfortunately, as you mentioned, this assumption is not possible to remove completely. To understand mathematically why this is the case (i.e., why the assumptions of the causal graph are necessary), consider an action $X \in \{0, 1\}$ and a reward $Y \in \{0, 1\}$ (unobserved). Without a very strong assumption that there is no latent confounding (which very rarely one can ascertain), we could always construct an SCM such that no policy $\pi(x)$ over $X$ could imitate the expert’s reward $E[Y]$. For concreteness, consider a structural causal model where $P(u) = P(x)$, $X \gets U$ and $Y \gets X \oplus \neg U$ where $\oplus$ is an “xor” operator. In this case,
> >$E[Y] = E[ U \oplus \neg U] = 1$,
>
> while for any $\pi(x)$,
> >$$
> E[Y|do(\pi)]
> $$
> $$
> = E[0 \oplus \neg U]\pi(X=0)+E[1 \oplus \neg U]\pi(X=1)
> $$
> $$
> = P(X=0)\pi(X=0) + P(X=1)\pi(X=1)
> $$
> $$
> < (P(X=0) + P(X=1))\times (\pi(X=0) + \pi(X=1))
> $$
> $$
> = 1.
> $$
>
> However, if a causal diagram $X \rightarrow Y$ is provided, it immediately follows from Theorem 2.1 that the imitator could always achieve the expert’s performance using an imitating policy $\pi(x) = P(x)$. In general, this example shows that the causal graph is indispensable for imitation learning, otherwise, the problem is underspecified. Given this necessity, a nice property of causal graphs is that they require the AI system to make these assumptions explicit and transparent, as opposed to other methods that keep them implicit, without much discussion or scrutiny.
>
> Having said that, there exist efficient methods that learn a set of candidate causal diagrams from the observational data. These methods have been studied under the rubrics of causal discovery, which is an ongoing subject of research (Spirtes, Glymour, and Scheines, 2001; Peters, Janzing, Scholkopft, 2017). Exploiting such methods in concert with our contribution would be an exciting topic of research. After all, we feel that our results are a necessary first step towards this, and towards a deeper understanding of imitation learning methods in challenging real-world applications.
>
>
>
> ---
> > _“the current analysis requires different covariates for each step, with no clear solutions for infinite (or long horizon) settings with some repeated structure, which seem to be the predominant setting in the real world.”_
>
> It is possible to apply our proposed methods to infinite horizon settings by combining other assumptions for temporal models. For example, one could divide the infinite horizon into multiple episodes; each episode contains a finite set of actions. Similar to dynamic Bayesian networks (Ghahramani, 1997), one could define a causal diagram representing a collection of invariant system dynamics that represent the transition between episode $t$ to $t+1$, for $t=1, 2, \dots$. We could then apply the sequential $\pi$-backdoor criterion and Algorithm 1 to find an imitating policy within a single episode. Note that such a policy is stationary with regard to different episodes. Repeatedly applying this policy for episode $t = 1, 2, \dots$ leads to an imitation strategy for infinite horizon settings. Overall, we acknowledge that causal imitation learning with an infinite horizon is an important research direction, which we hope to explore in the future.

---

> > ### Comment · Reviewer_1SfX · 2021-08-17
> > **Thanks to the authors for their for your response**
> >
> > Overall I am happy with the authors response. I wish they had also commented on my concern on the realism of the causal graphs used in the experiments, but I do realize that its a very nontrivial ask, and the contributions in the paper are already strong. I have updated my score and respond in more detail below.
> >
> > -----
> >
> > >_First, we note that oftentimes the reward signal  is unobserved and the reward function is not specified (Ng & Russell, 2000). In this case, Corollary 1 in (Zhang et al., 2020) showed that the expected reward  of any candidate policy is not identifiable_
> >
> > That makes sense, thanks for the reference. if we assume the reward signal is observable we begin to leave the IL regime and approach the RL regime, which while interesting is well outside the scope of this paper.
> >
> >
> > ---
> >
> > >Unfortunately, as you mentioned, this assumption is not possible to remove completely.
> >
> > Thank you for the example. I still believe that because the causal graph is so important yet also so difficult to obtain in practice, analysis on the effect of wrong causal graphs on causal imitation would be extremely valuable. That being said, its clear that the work in this paper is a crucial first step towards that goal.
> >
> > ---
> >
> > >It is possible to apply our proposed methods to infinite horizon settings by combining other assumptions for temporal models.
> >
> > Thanks for the initial ideas you've presented here, it would be great if they could make their way into the main paper!

---

### Official Review · Reviewer_nWYc · 2021-07-11

**Rating:** 7
**Confidence:** 1

**Summary:**

Paper presents a novel method to tackle the problem of causal imitation learning in a sequential setting. A graph-based criterion is proposed to measure the feasibility of causal imitation and an algorithm is derived to determine the imitability, which seems to be critical to the success of causal imitation learning. All introduced approaches come with theory and simulation experiments (on a simple toy domain).

**Main Review:**

My background is imitation learning in general and I have to admit I know very little about the technical details of causal learning (casual identification, confounding, etc). So I may not be able to access the technical contribution made by this paper on causal learning. But I do read this paper in detail twice and find it very illustrative and seems to be sound.

The empirical results can be limited -- only experiments on some toy domains are conducted. As I'm not familiar with the common practice in this topic, this may not be a serious issue.

From an imitation learning perspective, I do think the research problem here can be related to imitation learning from observation [1, 2, 3, 4] and over-imitation[5]. The authors are suggested to add citations to these papers.

[1] Behavioral Cloning from Observation

[2] Imitation Learning from Observations by Minimizing Inverse Dynamics Disagreement

[3] Third-Person Imitation Learning

[4] Imitation from Observation: Learning to Imitate Behaviors from Raw Video via Context Translation

[5] Mirroring without Overimitation: Learning Functionally Equivalent Manipulation Actions

**Time Spent Reviewing:**

2

---

> ### Author Response · Authors · 2021-08-10
> **Response to Reviewer nWYc**
>
> We appreciate the positive assessment of our paper and are glad that it was found to be critical to the success of causal imitation learning. We will study and add citations to the listed papers, thank you for sharing.
>
> ---
> > _“Empirical results can be limited”_
>
> We consider our contribution to be the formal and general understanding of which sets of variables to choose to observe when performing imitation learning. As such, the simulations were chosen to represent settings where ground-truth is available, and the performance of the new algorithm can be evaluated and compared against other methods. We hope that the experiments were convincing on two counts: 1) that the postulated bias is indeed present and affects other (non-causal) methods, and that 2) our approach does properly eliminate such biases. Using these results as a stepping stone, we are excited to explore applications of our approach in the context of general causal imitation learning in more challenging and realistic scenarios.

---

> > ### Comment · Reviewer_nWYc · 2021-08-21
> > **Response to the rebuttal**
> >
> > Thanks for the clarification and now I'm more affirmative on my recommendation.

---

### Official Review · Reviewer_3m9X · 2021-07-15

**Rating:** 7
**Confidence:** 3

**Summary:**

This paper investigates the problem of determining imitability in casual imitation learning in sequential settings. This is an extension from a previous work that studies the same problem in single-stage settings. The authors propose necessary and sufficient conditions for determining whether an imitator is able to unbiasedly imitate a demonstrator given a causal diagram. The authors further develop an efficient algorithm that outputs 1) whether such conditions are met and 2) a set of observable variables as inputs to the imitator policies to unbiasedly imitates the demonstrator. In addition, the authors demonstrate the performance of the algorithm and the usefulness of the conditions on both synthetic and real-world data.


**Limitations And Societal Impact:**

yes

**Main Review:**

The paper contributes to the foundations of imitation learning and attempts to answer an important question of whether an imitator is able to imitate a demonstrator if there exists sensor input mismatch. The results of the paper are applicable to all causal diagrams beyond Markov Decision Processes (MDPs). I think such fundamental research largely increases the practicality of imitation learning to fields that require formal guarantees, and provides a promising direction to understanding imitation learning.

However, I have some concerns of the practicality of the results on high-dimensional and complex problems. For example, t
he conditions proposed can determine whether a demonstrator can be imitated unbiasedly, but they do not answer other practical questions such as (1) in the biased case what variables should be used, and (2) given a finite amount of data, whether the set of variables found by Algorithm 1 is always better than other set of variables that do not guarantee imitability. But I think this paper is a good start to address these questions.

Other comments:

The efficiency of the Algorithm 1. The authors mention that this avoids the exponential complexity due to the need of checking all backdoor admissible sets. What’s the computational efficiency of Algorithm 1 then if not exponential? The experiments do not demonstrate the efficiency of the proposed algorithm as the structures considered in the synthetic experiments are rather small.

I think the authors did a good job in making this paper accessible to readers who are not familiar with do-calculus. However, the exposition of the theorems and the algorithm is a bit abstract. Is it possible to have a running tangible example? In addition, it would be great if the motivating car example in Page 2 can be improved. It’s a bit confusing, especially before I read Section 1.1. In L44, “the learner from Fig.1a had a full view of the car’s surroundings”. Which node in Fig. 1a corresponds to the learner? Based on the text, X is the human driver. If X also represents the learner, X does not seem to have access to a full view of surroundings as it is only connected to F (“looking forward”) and H (“car horns”). Furthermore, the mean of E[Y|do(\pi)] was not explained. And there is no concrete meaning of X being 0 or 1.

The results in the paper hold for non-stationary policies, i.e. having a distinct policy at every time step. Are the results applicable to stationary policies and infinite horizon settings as non-stationary policies are more expensive to learn and maintain?

In L 118, the authors mentioned that besides a causal diagram, a topological ordering is required. Is that a total ordering of all variables as in the Order column in Table 1?


**Time Spent Reviewing:**

4 hours

---

> ### Author Response · Authors · 2021-08-10
> **Response to Reviewer 3m9X**
>
> We are glad that our work was found to be a promising direction for imitation learning. We think that a theoretical backing for imitability based on causal understanding of the domain is going to be critical for agents operating in partially observable environments. We tried our best to clarify the issues raised in your review, as listed below, and would be happy to provide further elaboration if you find suitable.
>
> ---
> > _“The conditions proposed can determine whether a demonstrator can be imitated unbiasedly, but they do not answer other practical questions such as (1) in the biased case what variables should be used, and (2) given a finite amount of data, whether the set of variables found by Algorithm 1 is always better than other set of variables that do not guarantee imitability.”_
>
> As you noted, we introduced a new theoretical framework and develop conditions and algorithms to understand, with mathematical precision, whether imitation is in principle possible, noting that this was not known until this work. We believe this has been achieved and is a major contribution to the new discipline of causal imitation learning. Regarding your first point regarding when our theory determines that a demonstrator cannot be imitated, our paper makes no additional claim other than warning that this is the case. In other words, this implies that no matter how much data is available for the learner, there are distributions for which it will never be able to mimic the expert. We strongly believe this is a fundamental and important step in the right direction.
>
> Regarding your second question, our algorithm returns one admissible set (if it exists) but does not make claims about optimality, just consistency. In other words, this means that in the limit of the sample size, this set has the capability of generating perfect imitability (it’s consistent in a statistical sense). However, it’s possible that there are other covariate sets that may contain attractive properties (finite-sample behavior, better costs, ethically more desirable), and so on. Given that now we know whether a set exists (i.e., we can “decide” imitability) and how to find at least one admissible set (in case imitability is allowed), we feel the questions you raise are significant and can be answered building on our results. We agree with your observations and will take them as an exciting area for future explorations.
>
> ---
> > _“What’s the computational efficiency of Algorithm 1 then if not exponential?”_
>
> Regarding the time complexity of Algorithm 1, it first loops over all vertices ($V$), and for each of those vertices, it checks for all of its children ($E$). Algorithm 1 then runs HasValidConditioning to find a c-component (a $|V|+|E|$ operation) and checks independence. The independence check can be done using an algorithm for d-separation (in linear time), resulting in time complexity of $\mathcal{O}(|V|(|V|+|E|))$ for Algorithm 1. We will make sure to mention this in the text since this is the main benefit of our algorithm as compared to raw enumeration that takes exponential time.
>
> ---
> > _“it would be great if the motivating car example in Page 2 can be improved. It’s a bit confusing, especially before I read Section 1.1. In L44, “the learner from Fig.1a had a full view of the car’s surroundings”. Which node in Fig. 1a corresponds to the learner? Based on the text, X is the human driver. If X also represents the learner, X does not seem to have access to a full view of surroundings as it is only connected to F (“looking forward”) and H (“car horns”). Furthermore, the mean of E[Y|do(\pi)] was not explained. And there is no concrete meaning of X being 0 or 1.”_
>
> - In Fig. 1(a), node X represents the action of the expert (e.g., accelerate or decelerate).
> - It is true that since “the expert could have a full view of surroundings”, arrows from F, B, S, H to node X could all potentially exist. However, in this example, it is sufficient for the expert to make the optimal decision on X only based on the front view F and the car horn H, despite the availability of other information (B, S). Therefore, Fig. 1(a) only contains arrows F -> X and H -> X to reflect the actual underlying parameters of the policy it currently uses.
> - We consistently use E[Y|do(\pi)] to represent the expected reward of the learner; while E[Y] represents the performance of the expert
>
> Finally, we are grateful to the suggestions on the specific areas of confusion. We will address them in the future version of the manuscript.
>
>
> ---
> > _“Are the results applicable to stationary policies and infinite horizon settings as non-stationary policies are more expensive to learn and maintain?”_
>
> In principle, it would be possible to apply the proposed methods to infinite horizon settings by combining other assumptions for temporal models. For example, one could divide the infinite horizon into multiple episodes; each episode contains a finite set of actions. Similar to dynamic Bayesian networks (Ghahramani, 1997), one could define a causal diagram representing a collection of invariant system dynamics that represent the transition between episode $t$ to $t+1$, for $t=1, 2, \dots$. One could then apply the sequential $\pi$-backdoor criterion (Def. 2.3) and Algorithm 1 to find an imitating policy within a single episode. Note that such a policy is stationary with regard to different episodes. Repeatedly applying this policy for episode $t = 1, 2, \dots$ leads to an imitation strategy for infinite horizon settings. Having said that, we acknowledge that causal imitation learning with an infinite horizon is a challenging and significant research problem, which we hope can be explored in the future.
>
> ---
> > _“Is the order in Table 1 the topological ordering required by the approach?”_
>
> Yes, this is correct. We will add a comment to the figure description to clarify this point, thanks.

---

> > ### Comment · Reviewer_3m9X · 2021-08-22
> > **Thank the authors for the response**
> >
> > After reading authors' response and other reviewers, I would like to increase my score. The authors have nicely answered my questions and I think the work is indeed a solid step towards formal understanding of imitation learning.

---

### Official Review · Reviewer_4Bw1 · 2021-07-15

**Rating:** 7
**Confidence:** 1

**Summary:**

The paper addresses the task of behavioral cloning for sequential decision making problems, in a setting where the expert and the imitation learner have different sets of observable variables, i.e. in the presence of observational mismatch. To do this, the paper assumes knowledge of the causal graph that models the process used to generate the data. Using the graph, the authors characterize conditions under which imitation is possible and if it is, propose a process for coming up with the right subset of variables to use.

**Limitations And Societal Impact:**

The adequately discuss the limitations and societal impact of their work.

**Main Review:**

The paper is original and high quality. It is hard to parse for readers more familiar with imitation learning than causality, but that is inevitable to an extent. The main contribution is significant.

As a practitioner, I have the following questions.
1. In many practical deployments, people don't use behavioral cloning but instead deploy adversarial imitation learning, which has better properties than behavioral cloning when expert data isn't abundant. Can you share your thoughts whether it makes sense to match distributions over the subset of state features selected by your algorithm?
2. More broadly, a crucial condition for the applicability of the proposed method in practice is the amount of expert data it requires. Would you expect the method to work for a small dataset (e.g. just one expert trajectory)?
3. How would you approach the scenario where observations are images? It seems hard to impose causal structure on parts of an image and yet you use images as motivation in the description of your Figure 1a.

Minor point: Figure 4 has no confidence bars.


**Time Spent Reviewing:**

5

---

> ### Author Response · Authors · 2021-08-10
> **Response to Reviewer 4Bw1**
>
> We’re glad that the paper was found to be original and high quality, thank you. We agree that the notation can be rather dense at times. We expect to use the additional one page, if the paper is accepted, to expand the examples to further explain the intuition and reasoning behind the most critical parts of math. Please, find below the answers for the major issues raised in the review, thank you.
>
> ---
> > _“In many practical deployments, people don't use behavioral cloning but instead deploy adversarial imitation learning, which has better properties than behavioral cloning when expert data isn't abundant. Can you share your thoughts whether it makes sense to match distributions over the subset of state features selected by your algorithm?”_
>
> Yes, one could obtain an imitating policy by applying the adversarial imitation learning (AML) following an reverse order over actions $X$. More specifically, for $i = n, n-1, \dots, 1$, one could AML to train a decision rule $\pi_i(x_i|z_i)$ to match the observational distribution $P(x_i, z_i) = P(x_i, z_i | do(\pi_i))$. It follows from the properties of the sequential $\pi$-backdoor (Def. 2.3) that the resulting $\pi$ is an imitating policy. On the other hand, if no $\pi$-backdoor admissible set exists, matching the observational distributions will not lead to a valid imitation policy.  This is a great observation, thank you! We will incorporate it in the future version of the manuscript.
>
> ---
> > _“Would you expect the method to work for a small dataset?”_
>
> Our method is not data-dependent. Taken a causal graph as input, our method outputs a set of covariates to use with your chosen algorithm, so we see it as being complementary to methods that focus on specific use cases, such as small datasets. In other words, once a sequence of backdoor admissible covariates are found (regardless of the types of variables and numbers of samples), one could then apply state-of-art estimation methods (using such covariates as input) to learn an imitating policy from a small dataset. If our results do not apply, even with infinite samples the imitator will not succeed. Building on our results, we feel an important research direction will be to understand and characterize the finite-samples behavior of the algorithm for the positive instances (i.e., imitable) described by the theory developed here.
>
> ---
> > _“How would you approach the scenario where observations are images?”_
>
> There would need to be a pre-processing step which extracted objects from the image - for example, finding the locations and trajectories of all other vehicles in the surroundings, and treating those as variables. One could then apply causal discovery algorithms to learn a set of possible causal diagrams from the annotated, observational data. Causal discovery methods for learning causal relationships from data is a line of ongoing research  (Spirtes, Glymour, and Scheines, 2001; Peters, Janzing, Scholkopft, 2017). Overall, we consider our contribution to be a general and formal understanding of the precise conditions under which imitation is possible, which is a necessary step within a larger framework of causal imitation learning. We look forward to exploiting these insights for practical application in future work.

---

> > ### Comment · Reviewer_4Bw1 · 2021-08-20
> > **Response to rebuttal.**
> >
> > Thanks for the clarifications - they are really helpful.

---

### Official Review · Reviewer_rfa1 · 2021-07-30

**Rating:** 9
**Confidence:** 4

**Summary:**

The paper proposes to study causal imitation learning in a sequential framework by taking advantage of the structure of the causal graph. The authors present a necessary and sufficient criterion, distinct from the sequential backdoor criterion, in order to allow the feasibility of a causal imitation learning algorithm. Finally, the authors propose an algorithm to make it possible to define whether such a criterion is respected.

**Limitations And Societal Impact:**

The authors clearly discuss the limitations and societal impacts.

**Main Review:**

Strengths
========

The paper is clear, rigorous, and well written. Several examples inserted in the text facilitate the understanding and development of ideas. The paper clearly explains its theoretical limitations while addressing an important problem with concrete applications in ML.

Weaknesses
==========

The notation may seem heavy at first glance and requires special attention from readers to deeply understand the article. On the other hand, this notation seems necessary and is directly related to the rigor of the article. There is no free lunch.

Clarity
======

* *line 51*: $\mathbb{E} [y \mid \text{do}(\pi)]$: this expression has not yet been presented in the text.
* *line 106*: It would be clearer to use the expression *ch*.
* *line 122*: We think that the reader could benefit from making the link between the policy and the SCM explicit.
* *line 138*: It seems more natural to us to use the expression "*we have*" instead of "*such that*" from a logical perspective.
* Figure 3: Figure 3: faded edges are not defined.
* Table 1: If there is enough space, add some comments so as not to refer directly to the text. For example, what do the statistics in red mean?
* *line 217*: The term *ancestral graph* is not defined.
* Figure 4: The abbreviation $\pi-\text{BD}$ is not defined. Also, it would be preferable to use another abbreviation so as not to confuse the reader with Approach 3 presented in line 324.

Typos
=====

* fig. 1a *(line 39)*: Fig. 1a

**Time Spent Reviewing:**

6

---

> ### Author Response · Authors · 2021-08-10
> **Response to Reviewer rfa1**
>
> Thank you for your time and the positive assessment of our work. We certainly agree that this is a significant problem at the intersection of causality and imitation learning, and are glad that it was found to be mathematically rigorous and interesting.
>
> ---
> > _“The notation may seem heavy at first glance, and requires special attention [....] to deeply understand the article”_
>
> We agree that the paper can have rather dense notation at times. As mentioned in the review, some of this is unavoidable due to the technical nature of our contribution, but we are committed to making the text as clear as possible. We will incorporate the included suggestions, and attempt to expand on our examples to further explain the intuition using the additional extra page (if the paper is accepted).
>
> ---
> > _“Table 1: [...] For example, what do the statistics in red mean?”_
>
> Red indicates that (1) the imitation is not successful, i.e., the imitation error $|E[Y] - E[Y|do(\pi)]|$ is larger than a threshold; or (2) the expert’s performance is not imitable.

---

### Decision · Program_Chairs · 2021-09-28

**Decision:**

Accept (Oral)

**Comment:**

It is my pleasure to recommend acceptance of this paper.

As noted by reviewers:

*The paper is clear, rigorous, and well written. Several examples inserted in the text facilitate the understanding and development of ideas. The paper clearly explains its theoretical limitations while addressing an important problem with concrete applications in ML.*

Also:

*The paper contributes to the foundations of imitation learning and attempts to answer an important question of whether an imitator is able to imitate a demonstrator if there exists sensor input mismatch.*

**Consistency Experiment:**

NeurIPS has a long history of experimentation. In 2014, NeurIPS ran an experiment in which 10% of submissions were reviewed by two independent committees to quantify the randomness in the review process. This year, we repeated a variant of this experiment to see how the quality of the review process has changed over time.  This paper was part of the experiment and was therefore assigned to two committees (consisting of reviewers, an Area Chair, and a Senior Area Chair) that reached independent decisions.  If both committees made the same recommendation, this recommendation was followed. If a single committee recommended acceptance, the paper was accepted (with the exception of a few cases in which the other committee identified what we considered a fatal flaw, e.g., an error in a key result).

This copy’s committee reached the following decision: **Accept (Oral)**

The other committee assigned to the paper recommended **Accept (Poster)**.  You can find the other set of reviews, along with any follow up discussion with the authors here:
https://openreview.net/forum?id=Kvb0482Ysaf